# ATTACKING COMBINATORIAL BANDITS: BEYOND BOUNDED REWARDS

## ABSTRACT

Combinatorial Multi-Armed Bandits (CMABs) are a widely adopted tool to address online learning problems with a combinatorial nature. Adversarial attacks, on the other hand, represent a significant threat to machine learning algorithms, where a malicious entity intentionally manipulates data or feedback to deceive learning algorithms, undermining their performance and reliability. While CMABs and adversarial machine learning received extensive attention as distinct subjects, CMABs under adversarial attacks are still underinvestigated. We propose algorithms to attack CMABs, providing theoretical guarantees regarding success and cost in three different scenarios. Each scenario differs in the assumptions on the rewards. First, we study attacks when rewards are bounded and means are positive. Then, we consider two extensions in which rewards have unbounded support, distinguishing between positive and arbitrary means. For each scenario, we design two attack strategies. First, we assume that the attacker is omniscient, i.e., knows the problem instance, then we extend the attack to a more realistic setting where the learner and the attacker have the same knowledge of the problem. We show that our attack strategies are successful, i.e., the learner will select a target superarm for $T - o(T)$ times, except for some degenerate cases. We also show that in most settings the attack cost is sublinear in $T$. Finally, we validate our theoretical results via numerical experiments on synthetic instances.

## 1 INTRODUCTION

With the growing adoption of artificial intelligence systems, there is a rising interest in understanding their potential limitations and vulnerabilities in the face of malicious entities. We commonly refer to adversarial machine learning as the branch studying how to corrupt a system to alter its behaviour and how to design robust techniques resilient to malicious attacks. An attack is a precisely crafted noise that a malicious entity injects into the dataset used to train an algorithm or in the feedback received by an intelligent agent. The adversarial attack problem, initially studied in deep learning literature (Goodfellow et al., 2014; Sun et al., 2020; Inkawhich et al., 2019), also emerged in the decision-making area concerning reinforcement learning literature (Chen et al., 2019; Zhang et al., 2020; Pattanaik et al., 2017) and multi-armed bandits (Jun et al., 2018; Liu and Shroff, 2019; Wang et al., 2022).

Combinatorial bandits are an extension of the classic multi-armed bandit problem, where a learner interacts with an environment and chooses a combination of actions for each round Cesa-Bianchi and Lugosi (2012). Due to the combinatorial nature of several real-world problems, combinatorial bandits are widely used in practical applications Chen et al. (2013). Given the recent studies regarding attackability of MABs (Jun et al., 2018; Liu and Shroff, 2019; Wang et al., 2022), it is natural to study whether such vulnerabilities apply also to combinatorial bandits and wether it is possible to craft attack techniques to take advantage of the vulnerabilities of existing combinatorial bandits algorithms.

In the adversarial attack problem on multi-armed bandits, an attacker sits between the learner and the environment. The attacker designates a specific arm as a target and, upon observing the arm selected by the learner and the reward generated by the environment in each round, corrupts the reward feedback of the learner. The corruption is precisely crafted to make the learner believe that the target arm is optimal. It has been proved in stochastic bandits that adversarial attacks can easily fool state-of-the-art algorithms (Jun et al., 2018). In particular, it has been shown that stochastic

bandits are intrinsically vulnerable to adversarial attacks, meaning it is always possible to deceive a regret minimizer and make it select the target for $T - o(T)$ times while the attacker pays $O(\log T)$ cost (Jun et al., 2018; Liu and Shroff, 2019). The attackability of bandits has also been studied for linear bandits, where Wang et al. (2022) show that it is not always possible to attack with $O(\log T)$ cost. Concerning combinatorial bandits, as far as we know, the work of Balasubramanian et al. (2024) represents the state-of-the-art regarding attackability. They show that, with a positive reward assumption, when the attacker is not aware of the combinatorial structure of the problem, the attack is unfeasible. This shows the intrinsic robustness of the combinatorial framework against adversarial attacks. However, it is an open problem to characterize attackability in settings with more general assumptions on the reward feedback.

**Original contribution** Our research question regards whether it is possible to characterize the attackability of CMAB outside the positive reward assumption. We positively answer this question by providing conditions on the combinatorial bandit structure and showing that when such conditions hold, independently of the attack strategy, the bandit instance is unattackable, or the attack requires a linear cost. Interestingly, combinatorial bandits are more resilient than standard MABs to adversarial attacks. Indeed, there are conditions that depend on (i) how superarms are structured and (ii) the assumptions about rewards or means, that identify whether it is possible to craft an adversarial attack. In our paper, we analyse three settings and provide conditions guaranteeing non-attackability for each setting.

We analyze three different scenarios: bounded rewards with bounded means, unbounded rewards with positive means and unbounded rewards with signed means $[-1, 1]$. We provide three conditions that characterize the attackability in each setting, showing why such distinction on rewards and means is necessary. Intuitively, all these settings are equivalent from the perspective of a learning algorithm, e.g., CUCB. However, each setting has different constraints on the attack. For instance, the bounded reward assumption forces the attacker to keep the corrupted reward above 0 (in order to induce feasible rewards), limiting the options of the attacker.

For each scenario, we design two attack strategies: an oracle attack, where the attacker knows the means of each arm, and a realistic attack, where the attacker does not know them. We show that in combinatorial bandits the attack strategy design, the success of the attack, and cost strongly depend on some property of the instance. Table 1 reports a summary of the bounds obtained regarding the successfulness and cost for the proposed algorithms depending on whether some specific conditions are satisfied.

Notably, our results, restricted to the positive reward scenario, match the analysis of Balasubramanian et al. (2024) and, in the degenerate case where the CMAB reduces to a non-combinatorial stochastic bandit (i.e., when all superarms are composed of a single basic arm) our results match the bounds regarding attackability for the classic stochastic bandit problem up to constants.

| Rewards | Means $\boldsymbol{\mu}$ | Oracle $N_S^*(T)$ | Oracle $C(T)$ | Real $N_S^*(T)$ | Real $C(T)$ | Cond. |
|---|---|---|---|---|---|---|
| | | NA | NA | NA | NA | $C_1$ |
| $[0,1]$ | $[0,1]^m$ | $T - o(T)$ | $O(T)$ | $T - o(T)$ | $O(T)$ | $\neg C_1 \wedge C_2$ |
| | | $T - o(T)$ | $O(\log T)$ | $T - o(T)$ | $O(\log T)$ | $\neg C_1 \wedge \neg C_2$ |
| $\mathbb{R}$ | $[0,1]^m$ | $T - o(T)$ | $O(\log T)$ | $T - o(T)$ | $O(\log T)$ | |
| $\mathbb{R}$ | $[-1,1]^m$ | $T - o(T)$ | $O(T)$ | $T - o(T)$ | $O(T)$ | $C_3$ |
| | | $T - o(T)$ | $O(\log T)$ | $T - o(T)$ | $O(\log T)$ | $\neg C_3$ |

**Table 1:** The table summarizes our analysis in each proposed scenario. For each assumption on rewards and means we provide two type of techniques an oracle and a realistic attack. For each technique, we show number of times a learner selects the target arm $N_{S^*}(T)$ (i.e., the successful condition) and the cost $C(T)$. We mark with "NA" when the attack is not possible regardless the strategy used by the attacker. The column Cond. partitions each scenario showing the best bounds achievable given that the specified conditions are true or false. Condition $C_1$, $C_2$, and $C_3$ are provided in Condition 3.1, Condition 3.2, and Condition 3.3, respectively.

**Related works** The first work addressing the adversarial attack problem originated from stochastic bandits. The seminal works of Jun et al. (2018); Liu and Shroff (2019) provide the first analysis of attack strategies for the strong attack setting, where the attacker receives both the played arm and reward as feedback. Other works address the adversarial attack problem in adversarial bandits (Ma and Zhou, 2023; Yang et al., 2021), Gaussian process bandits (Bogunovic et al., 2020a; Han and Scarlett, 2022), and contextual bandits (Garcelon et al., 2020; Bogunovic et al., 2020b; Wang et al., 2022). Another parallel line of research concerns the analysis of how to defend from an attack. However, the majority of robust techniques focuses on a different setting where the attacker can only observe the reward vector: Lykouris et al. (2018) propose a robust version of the active arm elimination algorithm for stochastic bandits Gupta et al. (2019) design a robust algorithm agnostic to corruption, and Zhong et al. (2021) propose probabilistic sequential shrinking. Guan et al. (2020) design robust algorithm when the attack happens with a certain probability, and Rangi et al. (2022) propose a setting where the learner can access limited corruption-free samples. Concerning combinatorial bandits, Balasubramanian et al. (2024) propose a study on the attackability of combinatorial bandits with probabilistic triggered arms. In particular, they provide a condition under which the CMAB is attackable. In this paper, we focus on classical CMAB, extending their work along several directions. We provide a complete characterization of instances which are and are not attackable. Moreover, we consider more general CMAB and characterize the attackability on the basis of assumptions on rewards and means.

## 2 PRELIMINARIES

### 2.1 COMBINATORIAL FRAMEWORK

In a combinatorial multi-armed bandit (CMAB) problem a learner interacts with an environment composed of $m$ basic arms for $T$ rounds. At each time $t \in [T]$ [1] the learner can choose a combination of basic arm. Such combination is denoted as a superarm $S \in \mathcal{S}$ where $\mathcal{S} \subseteq 2^{[m]}$ defines the set of all possible superarms. Each basic arm $i \in [m]$ has a reward sampled i.i.d. from a $\sigma^2$-sub-Gaussian distribution. The vector $\boldsymbol{\mu} = \{\mu_1, \mu_2, \ldots, \mu_m\}$ includes the mean rewards of the arms. We denote with $d$ the largest cardinality of a superarm, i.e., $|S| \leq d \ \forall S \in \mathcal{S}$. In the following, $d$ will be an useful parameter to characterize the difficulty of the problem.

At each step $t \in T$, the environment samples an unbounded reward $X_{i,t} \in \mathbb{R} \ \forall i \in [m]$ and the learner, after selecting a superarm $S_t$, collects a total reward of $X_{S_t}$ with semi-bandit feedback. For the sake of clarity, we consider the reward to be a linear function

$$X_{S_t} = \sum_{i \in S_t} X_{i,t}$$

following the original CMAB problem described by Cesa-Bianchi and Lugosi (2012).

The objective of the learner is to minimize the regret, defined as:

$$R(T) = T \max_{S \in \mathcal{S}} \sum_{i \in S} \mu_i - \sum_{t=1}^{T} \sum_{i \in S_t} \mu_i \tag{1}$$

We denote the empirical mean of a basic arm at time $t$ as $\hat{\mu}_i(t)$. Given a superarm $S$, we denote the lowest mean basic arm as $\mu_S^* = \min_{i \in S} \mu_i$. $N_i(t)$ represents the number of times basic arm $i \in [m]$ has been selected in any superarm up to round $t$. Similarly, we write $N_S(t)$ to denote the number of times a particular superarm $S$ has been selected up to round $t$.

### 2.2 ADVERSARIAL ATTACK FRAMEWORK

The adversarial attack framework was originally proposed for stochastic multi-armed bandits by Jun et al. (2018) where an attacker sits in between the learner and environment and is able to manipulate the reward produced by the environment to trick the learner into selecting a specific arm rather than the optimal one. In CMAB the attacker, at each time $t \in [T]$ observes the superarm $S_t$ selected by the learner and the reward $X_{S_t} = \sum_{i \in S_t} X_{i,t}$ sampled by the environment.

---

[1] In this work, we refer as $[A]$, $A \in \mathbb{N}$, to the set $\{1, \ldots, A\}$.

---

**Algorithm 1** Interaction between learner, attacker and environment

---

    **for** $t \in T$ **do**
        Learner selects $S_t$
        Environment $\to X_{S_t} = \{X_{i,t} \; \forall i \in S_t\}$
        Attacker $\leftarrow (S_t, X_{S_t})$
        Attacker computes $A_{S_t} = \{A_{i,t} \; \forall i \in S_t\}$
        Learner $\leftarrow \hat{X}_{S_t} = X_{S_t} + A_{S_t}$
    **end for**

---

The attacker uses these information to craft a malicious reward $\hat{X}_t = X_t + A_t$ to lure the learner believing a target superarm $S^*$ optimal. Defining an attack algorithm means computing for each $t \in [T]$ a particular corruption $A_{S_t} = \sum_{i \in S_t} A_{i,t}$ to successfully lure the learner into selecting the target arm for the majority of the time. Algorithm 1 describes the interaction between the attacker, learner and environment. The attacker pays a cost equal to the total corruption injected $C(T) = \sum_{t=1}^{T} |A_{S_t}|$. The attacker succeeds when the learner does not select the target superarm $S^*$ a sublinear number of times paying a logarithmic cost. Formally, an attack algorithm is considered successful if it makes the learner pull the target super arm $S^*$ for $N_{S^*}(T) = T - o(T)$ times paying a $O(\log T)$ cost.

We analyse three different scenarios depending on the assumptions on the mean $\boldsymbol{\mu}$ and reward $X_S$: (i) bounded rewards in $[0,1]$ with positive means $\boldsymbol{\mu} \in [0,1]^m$, (ii) positive means $\boldsymbol{\mu} \in [0,1]^m$ and unbounded reward in $\mathbb{R}$, and (iii) signed means $\boldsymbol{\mu} \in [-1,1]^m$ and unbounded reward $\mathbb{R}$. For the sake of clarity, we normalize the means to be in $[0,1]$ and $[-1,1]$ but we remark that proposed results can be generalized to any range. While the three different scenarios can be address by the same learning algorithm (e.g., it is easy to see that Combinatorial UCB works in all scenarios), such scenarios are different from the perspective of the attacker. Indeed, each setting has different requirements on the attack, and, as we will show later, each scenario requires its own conditions on the attackable instances, and poses different challenges to craft an attack strategy. For instance, the bounded reward assumption forces the attacker to keep the corrupted reward above 0 (in order to induce feasible rewards), limiting the options of the attacker. This is the model studied by Balasubramanian et al. (2024). Here, we study also the setting with unbounded rewards, providing more stringent conditions on the attackability of the attack. We show that in all scenarios, with the exception of some degenerate cases, the attack is possible and can be crafted with a logarithmic cost. We provide two different attack strategies: an oracle attack and realistic attack. As customary in the adversarial attack literature we begin our analysis with an ideal scenario where the attacker knows the problem instance $\boldsymbol{\mu}$ usually denoted as oracle attack. Then, we study attacks on a real scenario where the attacker has the same knowledge of the learner, i.e., also the attacker needs estimates of $\boldsymbol{\mu}$.

Notice that, while Balasubramanian et al. (2024) consider the extension of CMAB with probabilistic triggered arms, their hardness example is analyzed in the classic CMAB problem which is a particular case of the probabilistic triggered setting. We generalize that result, providing a complete characterization of hard instances.

To simplify the exposition, in our theoretical analysis we will consider a learner implementing the Combinatorial Upper Confidence Bound (CUCB) algorithm. However, our results only employ the no-regret property of the regret minimizer, and in particular that it provides $O(\log T)$ regret. Hence, our results can be generalized to any learner that satisfies this assumption.

## 3 ORACLE ATTACK STRATEGIES

In this section, we consider the problem faced by an attacker that knowing the mean rewards $\mu$. In classical MABs, Jun et al. (2018) provides a straightforward attack that force the learner to select a target arm. In particular, it is sufficient to corrupt every arm different than the target. In CMABs, it is not straightforward how to define an equivalent attack,. Indeed, even knowing $\mu$ there can be several ways to corrupt every arm $i \in S$. Moreover, other challenges derive from the CMAB framework itself and may lead to degenerate cases making the attack impossible. For instance, consider the case where the target $S^* \subset S$ where $S$ is a generic superarm. In this setting, corrupting every basic arm $i \in S$ will also affect the estimate of the target superarm. Furthermore, depending on the assumptions

about the rewards and problem instance we can have degenerate cases where the attack is not possible. To ensure that the learner believes the target $S^*$ optimal, we design the attack such that every other superarm $S \in \mathcal{S} s.t. S \neq S^*$ lead to a worse reward. At each time $t$, our attack will guarantee that when the learner selects $S_t$ the proposed oracle attack strategy corrupts *equally* every basic arm $i \in [m]$ s.t. $i \in S_t \wedge i \notin S^*$ reducing the estimate of every arm $i$ to be $\mu_i \leq \mu^*$ where $\mu_{S^*}^*$ is the lowest mean basic arm in $S^*$. Now, we propose an attack for each setting highlighting the possible degenerate conditions.

**Bounded rewards, positive means**    When rewards $X_{i,t} \in [0,1] \, \forall i \in [m], \forall t \in [T]$ and means $\boldsymbol{\mu} \in [0,1]^m$, the attacker cannot output a negative reward to the learner. Indeed, this would contradict the assumption on positive rewards, alarming the learner. This limitation can be problematic depending on how super arms are structured. In the following, we identify two conditions that determine whether the attacker, independently from its strategy, cannot make the learner prefer the target $S^*$ over a generic arm $S$. Next, we show that this conditions are "tight", i.e., that an attack is possible if both the conditions are *not* satisfied.

**Condition 3.1** (C1). *There exists an $S \in \mathcal{S}$ with $S^* \subset S$.*

In the following proposition, we show that if the combinatorial structure of superarms satisfies Condition 3.1, then no attack can be successful in this scenario.

**Proposition 3.1.** *If Condition 3.1 is satisfied, in the bounded rewards, positive means setting no attack can be successful.*

Intuitively, no attack is possible since even decreasing the estimate of every basic arm $i \notin S^*$ to 0, the learner would be indifferent between $S$ and $S^*$. A CUCB leaner would still prefer $S$ to $S^*$ because of its bigger exploration term linked to the arms not included in $S^*$.[2]

Condition 3.2 highlights another set of instances that are hard to attack. In particular, it encompass some of the instances where the $S^*$ superarm presents a basic arm with null mean and a superarm $S$ is included in $S^*$.

**Condition 3.2** (C2). *There exists an arm $i \in S^*$ with $\mu_i = 0$ and an $S \subset S^*$ such for all $j \in S^*$, $\mu_j \neq 0$, it holds $j \in S$.*

Intuitively, if 3.2 is satisfied, then there exists a superarm that is included in $S^*$ and has the same expected reward of $S^*$. The following proposition shows that in this setting it is impossible to reduce the expected reward of $S$, and the only way to craft an attack is to increase the reward of $S^*$.

**Proposition 3.2.** *If Condition 3.2 is satisfied, in the bounded rewards, positive means setting all successful attacks requires a positive corruption on the target superarm $S^*$.*

Despite this negative result, Condition 3.2 is a less challenging scenario than Condition 3.1. Indeed, as we will show in the following an attack is possible but requires the attacker to pay a linear cost. Intuitively, Proposition 3.2 shows that even bringing all non-target basic arms to an empirical mean of 0 would make the superarms $S$ and $S^*$ equivalent for what concerns the means. Therefore, a learner will be indifferent between the two arms and the attack would not be successful. We will be able to make the attack successful corrupting the basic arms in $S^*$. However, in such a case the cost would be linear for the attacker.

Now, that we provide the two negative conditions, we are ready to design our attack. Our attack is successful whenever Condition 3.1 and Condition 3.2 do not hold. In particular, we propose the following attack strategy:

$$A_{i,t} = \begin{cases} -X_{i,t} & \text{if } i \in S_t \wedge i \notin S^* \\ 0 & \text{otherwise} \end{cases} \tag{2}$$

The following proposition shows that the Attack strategy in Eq. (2) is successful and that the attacker pays a $O(\log T)$ cost.

**Proposition 3.3.** *Consider a CMAB problem with bounded rewards and positive means. Assume the learner experience $O(\log T)$ regret, and the oracle attacker follows the attack strategy in Eq. (2). Then,*

---

[2]Notice that the previous result can be proven even for other regret minimizers. Indeed, even in the best-case scenario, a regret minimizer would be indifferent between superarm $S$ and $S^*$, playing $S^*$ half of the times.

*if Condition 3.1 and Condition 3.2 do not hold, the attack is successful i.e., $\mathbb{E}[N_{S^*}(T)] = T - o(T)$ with cost $O(\log T)$.*

Notice that no attack can solve the structural issue related to Condition 3.1. However, if the attacker is willing to pay linear cost, Condition 3.2 can be mitigated corrupting with a small positive constant $\epsilon > 0$ every arm $i \in S^*$ s.t. $\mu_i = 0$ as shown in Eq. 3.

$$A_{i,t}^{\epsilon} = \begin{cases} -X_{i,t} & \text{if } i \in S_t \wedge i \notin S^* \\ \epsilon & \text{if } i \in S_t \wedge i \in S^* \wedge \mu_i = 0 \\ 0 & \text{otherwise} \end{cases} \tag{3}$$

However, this is possible only if the attacker has knowledge of the true means $\boldsymbol{\mu}$.

**Unbounded rewards, positive means**  Differently from the previous setting, since rewards can now take values in $\mathbb{R}$, and means $\boldsymbol{\mu} \in [0,1]^m$, Conditions 3.1 and 3.2 do not present a problem as the attacker can induce a negative reward to the learner. The attack in Eq. 4 exploits a fixed corruption due to the knowledge of $\boldsymbol{\mu}$.

$$A_{i,t}^{\epsilon} = \begin{cases} -(\mu_i + \epsilon) & \text{if } i \in S_t \wedge i \notin S^* \\ 0 & \text{otherwise} \end{cases} \tag{4}$$

for some arbitrary small parameter $\epsilon > 0$. The following proposition shows that attack in Eq. (4) is successful and attacker pays logarithmic cost.

**Proposition 3.4.** *Consider a CMAB problem with unbounded rewards and positive means. Assume the learner experience $O(\log T)$ regret, and the oracle attacker follows the attack strategy in Eq. (4). Then, the attack is successful i.e., $\mathbb{E}[N_{S^*}(T)] = T - o(T)$ with cost $O(\log T)$.*

Notice that attack in Eq. (4) does not rely on the realization of the basic arm as in Eq. (2) and shifts the original problem to another one where there are no superarms that are better choices than the target. Shifting the distribution (and keeping noise rewards) make the attack *stealth* meaning that is also more difficult to detect.

**Unbounded rewards, signed means**  When rewards are unbounded $\mathbb{R}$ with signed means $\boldsymbol{\mu} \in [-1,1]^m$, Conditions 3.1 and 3.2 are not problematic anymore. However, we identify another structural issue:

**Condition 3.3** (C3). *There exists an $S \subset S^*$ such that $\mathbf{E}[X_S] \geq \mathbf{E}[X_{S^*}]$.*

Then, according to the following Proposition 3.5:

**Proposition 3.5.** *Suppose that Condition 3.3 is satisfied. Then, any successful attack has a linear cost.*

Intuitively, if we attack only the arms $i \notin S^*$ as we did up to now, the superarms composed of only the basic arms with positive mean $S = \{i : \mu_i > 0 \wedge i \in S^*\}$ which are subset of the target $S \subset S^*$ would always be preferred to $S^*$. If Condition 3.3 is true, we can still achieve a successful attack corrupting arms $i \in S^*$ s.t. $\mu_i < 0$ with a positive corruption, but the attacker pays a linear cost. The attack is reported in Eq. (5) below.

$$A_{i,t}^{\epsilon} = \begin{cases} \max\{-\mu_i + \epsilon, 0\} & \text{if } i \in S_t \wedge i \in S^* \\ -\max\{\mu_i, 0\} & \text{if } i \in S_t \wedge i \notin S^* \\ 0 & \text{otherwise} \end{cases} \tag{5}$$

for some small parameter $\epsilon > 0$. Proposition 3.6, state the successfulness and cost of attack in Eq. (5).

**Proposition 3.6.** *Consider a CMAB problem with unbounded rewards and signed means. Assume the learner experience $O(\log T)$ regret, and the oracle attacker follows the attack strategy in Eq. (5). Then, if Condition 3.3 holds, the attack is successful i.e., $\mathbb{E}[N_{S^*}(T)] = T - o(T)$ with linear cost $O(T)$.*

However, when Condition 3.3 does not hold, we can design a successful attack with logarithmic cost as follows:

$$A_{i,t}^{\epsilon} = \begin{cases} -\max\{\mu_i - \mu^* \mathbb{I}(\mu^* < 0) + \epsilon, 0\} & \text{if } i \in S_t \wedge i \notin S^* \\ 0 & \text{otherwise} \end{cases} \tag{6}$$

where $\mathbb{I}(\cdot)$ is the indicator function. As for previous cases, it is easy to see that the attack in 5 shifts the original problem to another one where the target superarm is the best choice for the learner. Indeed, if the basic arms in the target are the only ones with a positive mean, it is logical to say that any other combination of basic arms other than the target could only worsen the total expected reward.

## 4 REALISTIC ATTACK STRATEGY

In a realistic scenario the attacker has no information on the true means $\boldsymbol{\mu}$. Under this assumption the attacker must build an estimate $\hat{\mu}_i, \forall i \in [m]$ while attacking, to craft a precise corruption. To this extent, we define the event $E = \{|\hat{\mu}_i(t) - \mu_i| \leq \beta(N_i(t)) \quad \forall i, \forall t\}$ where given a probability $\delta > 0$, as proved by Lemma 4.1 the function $\beta(N)$, defined in Eq. (7) represents the confidence radius of an estimate $\hat{\mu}_i$:

$$\beta(N_i(t)) = \sqrt{\frac{\ln(\frac{2mT}{\delta})2\sigma^2}{N_i(t)}} \tag{7}$$

The function in Eq. (7) is decreasing in $N_i(t)$ and expresses the distance between the true mean and the correspondent estimator of basic arm $i$. Now, we are ready to show that $E$ holds with high probability.

**Lemma 4.1.** *For any $\delta \in (0,1), \mathbb{P}(E) > 1 - \delta$.*

Notice that the Conditions 3.1, 3.2, and 3.3 are valid regardless of the attacker's knowledge. Indeed, the limitations induced by the conditions concern the combinatorial structure of the superarms. Similarly to Section 3, we divide our analysis for the different assumptions about rewards and means. While we assume the bandit acts according to CUCB algorithm, it is easy to extend our techniques to be agnostic to the algorithm implemented by the learner, requiring only that the learner is a regret minimizer.

**Bounded rewards, positive means** Similarly to oracle attack, we start considering the setting in which rewards $X_{i,t} \in [0,1] \; \forall i \in [m]$ and $t \in [T]$, and means $\boldsymbol{\mu} \in [0,1]^m$. If Condition 3.1 or 3.2 are satisfied, the same problems arise. In the worst case Condition 3.1 holds, and the attack cannot be successful. However, when both Condition 3.1 and 3.2 are not satisfied, the best we can do is to attack relying on the realization of each basic arm. We can apply the same attack of Eq. (2) that is easy to implement, does not require knowledge of the means, is successful, and has logarithmic cost. Thus, we can use the same attack in Eq. (2) defined for the oracle attacker.

**Unbounded rewards, positive means** Now, suppose that rewards are in $\mathbb{R}$ and means $\boldsymbol{\mu} \in [0,1]^m$. Similarly to Section 3, there are no conditions tackling the attackability of this scenario. However, lacking the knowledge of the true means $\boldsymbol{\mu}$ the attacker must exploit its own estimates for the attack. The corruption begins the after $d$ rounds, when it is sure the learner has selected every basic arm at least once. In Definition 8 we propose a new attack tailored for this setting:

$$A_{i,t}^{\epsilon} = \begin{cases} -(\hat{\mu}_i(t) + \epsilon) & \text{if } i \in S_t \wedge i \notin S^* \\ 0 & \text{otherwise} \end{cases} \tag{8}$$

Where $\epsilon > 0$ is an arbitrary small parameter of the attacker and $\hat{\mu}_i(t)$ is the corruption free estimate of arm $i \in [m]$. The following proposition provides the guarantees of our attack.

**Proposition 4.2.** *Consider a CMAB problem with unbounded rewards and positive means. Assume the learner experiences $O(\log T)$ regret, and the attacker follows the attack strategy in Eq. (8). Then, the attack is successful, i.e., $\mathbb{E}[N_{S^*}(T)] = T - o(T)$ with cost $O(\log T)$.*

Proposition 4.2 shows that the successfulness and cost match the one for the oracle attack. Notice that similarly to the other results in the paper, this technique is not dependent by the algorithm used by the learner.

**Unbounded rewards, signed means**   Finally, we consider the most general case: rewards are unbounded and means $\boldsymbol{\mu} \in [-1, 1]$, with the only assumption that they follow a sub-Gaussian distribution. Similarly to 3, when Condition 3.3 holds a successful attack would need to make the empirical mean of the target basic arms all positive and all the other ones negative. Hence, we propose the following attack:

$$
A_{i,t}^{\epsilon} = \begin{cases} (\max\{-\hat{\mu}_i(t), 0\} + \beta(N_i(t)) + \epsilon) & \text{if } i \in S_t \wedge i \in S^* \\ -(max\{\hat{\mu}_i(t), 0\} + \beta(N_i(t)) + \epsilon) & \text{if } i \in S_t \wedge i \notin S^* \\ 0 & \text{otherwise} \end{cases}
\tag{9}
$$

The following proposition states the guarantees of our attack.

**Proposition 4.3.** *Consider a CMAB problem with unbounded rewards and signed means. Assume the learner experiences $O(\log T)$ regret, and the attacker follows the attack strategy in Eq. 9 then the attack is successful i.e., $\mathbb{E}[N_{S^*}(T)] = T - o(T)$ with cost $O(T)$.*

However, when Condition 3.3 is not satisfied, we can attack according as follows:

$$
A_{i,t}^{\epsilon} = \begin{cases} -(\max(\hat{\mu}_i(t) - \hat{\mu}^*(t), 0) + \beta(N_i(t)) + \beta(N_{i^*}(t)) + \epsilon) & \text{if } i \in S_t \wedge i \notin S^* \\ 0 & \text{otherwise} \end{cases}
\tag{10}
$$

where $\epsilon$ is an arbitrary small parameter. Then, Proposition 4.4 shows the successfulness and cost of Eq. (10).

**Proposition 4.4.** *Consider a CMAB problem with unbounded rewards and signed means. Assume the learner experiences $O(\log T)$ regret, and the attacker follows the attack strategy in Eq. 10. Then, the attack is successful i.e., $\mathbb{E}[N_{S^*}(T)] = T - o(T)$ with with cost $O(\log T)$.*

## 5 EXPERIMENTS

In this section, we provide numerical experiments to validate our theoretical claims. We consider a learner implementing CUCB algorithm and for each configuration of rewards and means, we attack following our proposed techniques. Figure 1 shows our results. For each setting, we evaluate the cumulative regret and the attack cost of the oracle attack and the realistic attack.

**Bounded rewards, positive means**   In this setting, we compare a CUCB learner against oracle attack defined in Eq. (2). We evaluate the case when Condition 3.1 and 3.2 do not hold, that is when the attacker is able to achieve a successful attack with $O(\log T)$ cost. We define $m = 4$ and $d = 2$. The means $\boldsymbol{\mu} = \{\mu_1, \mu_2, \mu_3, \mu_4\} = \{\Delta, \Delta, \Delta/2, \Delta/2\}$ with $\Delta = 0.5$. The target superarm $S^* = \{3, 4\}$ where the values represent the index of the correspondent basic arm $i \in [m]$. Furthermore, we set $\sigma = 0.1$. Time horizon $T = 10^5$ and we perform $E = 10$ experiments. For reproducibility, we set the random seed to 123. Since in this particular scenario, the oracle attack (OA) coincide with the realistic attack, we report only the former. Figures 1a and 1d show the cumulative regret and attack cost, respectively.

**Unbounded rewards, positive means**   In this setting, we compare a CUCB learner against oracle attack defined in Eq. (4) and the realistic attack in Eq. (8). In this scenario, the attack is successful and has logarithmic cost even if Conditions 3.1 and 3.2 hold. To show this behaviour, we define $m = 4$ and $d = 3$ to force the existence of a superarm $S \subset S^*$ creating the scenario described in Condition 3.1. The means $\boldsymbol{\mu} = \{\mu_1, \mu_2, \mu_3, \mu_4\} = \{\Delta, \Delta, \Delta/2, \Delta/2\}$ with $\Delta = 0.5$. The target superarm $S^* = \{3, 4\}$ where the values represent the index of the correspondent basic arm $i \in [m]$. Again, we set $\sigma = 0.1$. We set the attacker parameter $\epsilon = 0.01$ for both the oracle attacker and the realistic attacker. The time horizon is $T = 10^5$ and we perform $E = 10$ experiments with the random seed to 123. Figures 1b and 1e show the cumulative regret and attack cost, respectively.

**Unbounded rewards, signed means**   We compare a CUCB learner against the oracle attack defined in Eq. (5) and the realistic attack in Eq. (9). In this scenario, we report experiments for the more general case where Condition 3.3 is satisfied. As in the previous case, Conditions 3.1 and 3.2 do not alter successfulness and attack cost. We define $m = 4$ and $d = 2$. The means are

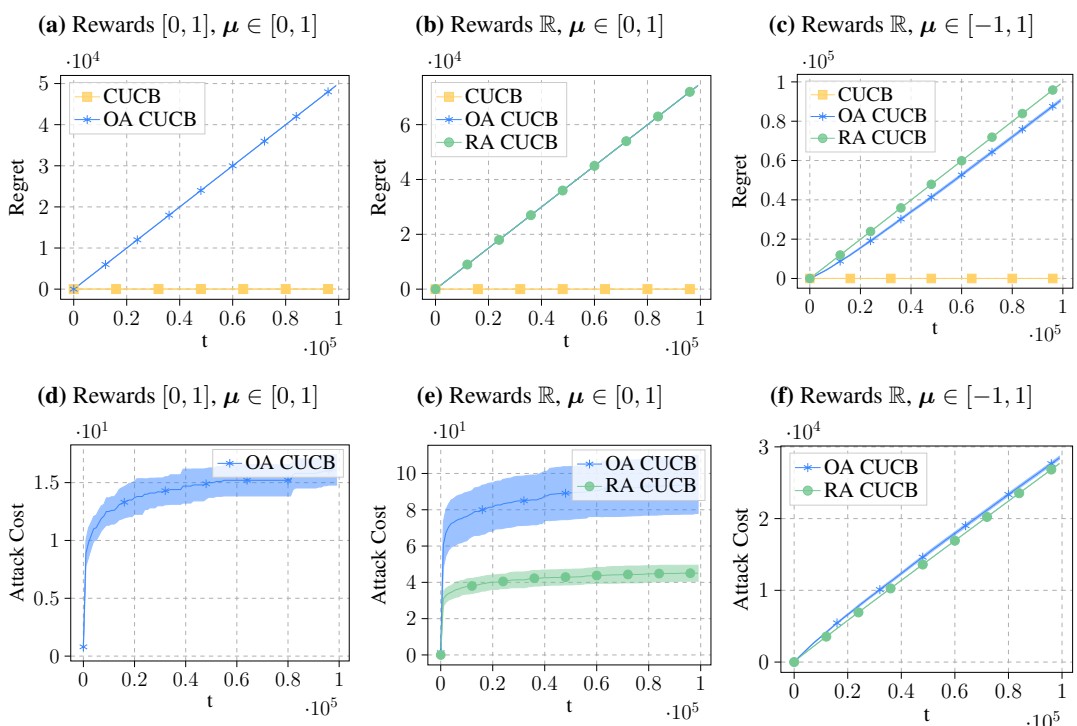

**Figure 1:** The figure summarizes the results of our numerical experiments. For each setting, we report the cumulative regret and attack comparing a non-corrupted CUCB learner and a corrupted CUCB learner against the oracle attacker (OA) and the realistic attacker (RA), except for the setting when rewards are bounded and means are positive since the strategies for OA and RA coincide. We evaluate each experiment with a $95\%$ confidence interval over $E = 10$ experiments. In particular, Figures 1a,1d refer to attack strategy in Eq. (2), Figures 1b and 1e refers to Eq. (4) and Eq. (8), and Figures 1c, 1f refers to Eq. (5) and Eq. (9).

$\boldsymbol{\mu} = \{\mu_1, \mu_2, \mu_3, \mu_4\} = \{\Delta, \Delta, \Delta/2, -\Delta/2\}$ with $\Delta = 0.5$. The target superarm $S^* = \{3, 4\}$, where the values represent the index of the correspondent basic arm $i \in [m]$. As before, we set $\sigma = 0.1$ and the attacker parameter $\epsilon = 0.01$ for both the oracle attacker and the realistic attacker. The time horizon is $T = 10^5$ and we perform $E = 10$ experiments with the random seed to $123$. Figures 1b and 1e show the cumulative regret and attack cost, respectively.

# 6 CONCLUSIONS

We study the attackability of combinatorial multi-armed bandits in different settings outside the positive mean assumption. We identified conditions regarding the intrinsic structure of the super arms that, when satisfied, make the current instance unattackable or require the attacker to pay linear cost. The conditions also depend on the assumptions about rewards and means of each basic arm. For this reason, we divided our analysis into different settings, considering bounded positive (or unbounded) rewards and positive (or signed) means. In each setting, identified by rewards and means assumptions, we propose an attack strategy (an oracle attack and a realistic attack) that matches the best bounds achievable concerning successfulness and attack cost, given the respective limitations captured by the structural conditions. Surprisingly, our attack strategies are valid regardless of the algorithm used by the learner. Finally, we provide numerical experiments to support our theoretical claims. Our work exposes several limitations in attacking combinatorial bandits, primarily depending on the intrinsic structure of the superarms.

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

# A OMITTED PROOFS

**Lemma 4.1.** *For any $\delta \in (0, 1), \mathbb{P}(E) > 1 - \delta$.*

*Proof.* Proving that $P(E) \geq 1 - \delta$ is equivalent to prove that $P(E^c) \leq \delta$ where $E^c$ is the complementary event. Now let

$$E_{i,t}^c = \{|\hat{\mu}_i(t) - \mu_i| \leq \beta(N_i(t))\},$$

then we have that:

$$P(E^c) = P\left(\bigcup_{i=1}^{K}\bigcup_{t=1}^{T} E_{i,t}^c\right)$$

$$\leq \sum_{i=1}^{K}\sum_{t=1}^{T} P\left(E_{i,t}^c\right) \tag{11}$$

$$\leq \sum_{i=1}^{K}\sum_{t=1}^{T} 2\exp\left\{-\frac{N_i(t)\beta(N_i(t))^2}{2\sigma^2}\right\} \tag{12}$$

$$\leq \delta, \tag{13}$$

where in Inequality (11) we applied the Union Bound, in Inequality (12) the Hoeffding Bound and Inequality (13) follows by substituting $\beta(N_i(t))$ defined in Equation (7). $\qquad\square$

## A.1 CONDITIONS

**Condition 3.1** (C1). *There exists an $S \in \mathcal{S}$ with $S^* \subset S$.*

**Proposition 3.1.** *If Condition 3.1 is satisfied, in the bounded rewards, positive means setting no attack can be successful.*

*Proof.* Let $S^*$ be the target arm choosen by the attacker. Consider any problem instance where is true that $\exists S$ s.t. $S^* \subset S$. This means that for a generic target super arm $S^* = \{a_1, \ldots, a_n\}$, exists a superarm $S$ such that $S = \{a_1, \ldots, a_n, b_1, \ldots, b_k\}$ for some $n, k \in \mathbb{N}$. Even if the attacker corrupts every arm $b_i$ with $i \in [k]$ to be 0, since every $a_j \geq 0, \forall j \in [n]$ and $b_i \geq 0, \forall i \in [k]$ is always true that $X_S \geq X_{S^*}$. It follows that the attacker, independently from the corruption injected cannot induce a preference over the target arm $S^*$. $\qquad\square$

**Condition 3.2** (C2). *There exists an arm $i \in S^*$ with $\mu_i = 0$ and an $S \subset S^*$ such for all $j \in S^*$, $\mu_j \neq 0$, it holds $j \in S$.*

**Proposition 3.2.** *If Condition 3.2 is satisfied, in the bounded rewards, positive means setting all successful attacks requires a positive corruption on the target superarm $S^*$.*

*Proof.* Let $S^*$ be the target arm chosen by the attacker. Consider any problem instance where is true that $\exists i \in S^*$ s.t. $\mu_i = 0$ and $\exists S$ s.t. $\forall j \in S$ with $j \neq i$ is true that $j \in S^*$. This is a specular condition of Condition 3.1. This means that for a generic target super arm $S^* = \{a_1, \ldots, a_n, b\}$ with $\mu_b = 0$, $\exists S$ such that $S = \{a_1, \ldots, a_n\}$ for some $n \in \mathbb{N}$. Then, if the attacker corrupts only $i \notin S^*$ will always be true that $X_S = X_S^*$, thus again the attacker cannot induce a preference order to the target $S^*$. $\qquad\square$

**Condition 3.3** (C3). *There exists an $S \subset S^*$ such that $\mathbf{E}[X_S] \geq \mathbf{E}[X_{S^*}]$.*

**Proposition 3.5.** *Suppose that Condition 3.3 is satisfied. Then, any successful attack has a linear cost.*

*Proof.* Let $S^*$ be the target arm chosen by the attacker. Consider any problem instance where is true that $\exists S$ s.t. $S \subset S^*$. This means that for a generic target super arm $S^* = \{a_1, \ldots, a_n, b_1, \ldots, b_k\}$, exists a superarm $S$ such that $S = \{a_1, \ldots, a_n\}$ for some $n, k \in \mathbb{N}$. Then, if $\sum_{b_i \in S^*} \mu_{b_i} < 0$ it is true that $X_S > X_{S^*}$. Therefore, to induce a preference over the target $S^*$ superarm, the attacker must add a positive corruption to each $b_i \in S^*$. Since in a successfull the $S^*$ is played $N_{S^*} = T - o(T)$ times, this makes the corruption cost linear. $\qquad\square$

## A.2 ORACLE ATTACKS

**Proposition 3.3.** *Consider a CMAB problem with bounded rewards and positive means. Assume the learner experience $O(\log T)$ regret, and the oracle attacker follows the attack strategy in Eq. (2). Then, if Condition 3.1 and Condition 3.2 do not hold, the attack is successful i.e., $\mathbb{E}[N_{S^*}(T)] = T - o(T)$ with cost $O(\log T)$.*

*Proof.* Since the attacker corrupts from $t = 0$ the learner face a different problem where only the target $S^*$ is the optimal arm. However, this is true only if Condition 3.1 and 3.2 are not true. Indeed the $S^*$ is optima only if the learner can induce a preference order with respect to other superarms $S$. All the instances where is not true Condition 3.1 and 3.2 are of type:

$$\mathcal{S}^{1,2} = S^* \setminus S \cap \{i : \mu_i = 0\} > 0 \neq \varnothing, \forall S.$$

Thus, suppose to bring every mean $\mu_i = 0$ $0 \forall i \notin S^*$ with instance of type $\mathcal{S}^{1,2}$, given that the total rewards of $S^*$ is $X_{S^*} = \sum_{i \in S^*} \mu_i$ it is true that:

$$X_S = \sum_{i \in S} \mu_i^c$$

$$= \sum_{i \in S^* \cap S} \mu_i$$

$$< \sum_{i \in S^* \cap S} \mu_i + \sum_{i \in S^* \setminus S} \mu_i$$

$$= \sum_{i \in S^*} \mu_i.$$

Thus we proved that in the instances of type $\mathcal{S}^{1,2}$, the attacker can induce a preference over the $S^*$. Since corruption starts at round $t = 0$ the $S^*$ is now the optimal arm in the corrupted problem that the learner is facing. Given that the learner is a $O(\log T)$ regret minimizer, is also true that there exists $D > 0$ such that:

$$\sum_{S \in \mathcal{S}} \mathbb{E} N_S(T) \left( X_{S^*} - X_S \right) \leq D \log T, \tag{14}$$

where $X_{S^*}$ and $X_S$ are the total exact means of superarms $S^*$ and S respectively as seen by the learner. Therefore, for any $S \neq S^*$, we have:

$$\mathbb{E} N_S(T) \left( X_{S^*} - X_S \right) \leq D_S \log T$$

$$\leq \frac{D_S}{X_{S^*} - X_S} \log T$$

$$= O(\log T). \tag{15}$$

Thanks to the result in Eq. (15), we deduce that a logarithmic regret bound implies that the bandit algorithm satisfies $\mathbb{E} N_S(T) = O(\log(T))$. That is, for a large enough $T$, $\mathbb{E} N_S(T) \leq D_S \log(T)$ for some $D_S > 0$. Based on the view that the oracle attack effectively shifts the means of the super-arms $X_S, S \in \mathcal{S}$, we recall that the best super-arm is $S^*$. Then:

$$\mathbb{E} N_{S^*}(T) = T - \sum_{S \neq S^*} \mathbb{E} N_S(T)$$

$$\geq T - \sum_{S \neq S^*} D_S \log T$$

$$= T - o(T),$$

which proves the first part of our statement. For the second statement, we notice that $\mathbb{E} N_S(T) = D_S \log T$ for any $S \neq S^*$, that we do not attack the $S^*$ super-arm and that the attack is limited to 1. Therefore,

$$\mathbb{E} \left[ \sum_{t=1}^{T} |A_{S_t}^\epsilon| \right] \leq \sum_{S \neq S^*} \mathbb{E} N_S(T) \cdot 1 \leq$$

$$\leq \sum_{S \neq S^*} D_S \log T =$$

$$= O\left( \log T \right).$$

$\square$

**Proposition 3.4.** *Consider a CMAB problem with unbounded rewards and positive means. Assume the learner experience $O(\log T)$ regret, and the oracle attacker follows the attack strategy in Eq. (4). Then, the attack is successful i.e., $\mathbb{E}[N_{S^*}(T)] = T - o(T)$ with cost $O(\log T)$.*

*Proof.* The proof of the first part of the proposition is identical to the proof of Proposition 3.3. For the second statement, we notice that $\mathbb{E}N_S(T) = D_S \log T$ for any $S \neq S^*$ and we recall that we do not attack $S^*$. Therefore, if attacker acts according to Eq. 4 the cost is:

$$\mathbb{E}\left[\sum_{t=1}^{T} |A_{S_t}^\epsilon|\right] = \sum_{S \neq S^*} \mathbb{E}N_S(T) \cdot A_S^\epsilon$$

$$\leq \sum_{S \neq S^*} D_S A_S^\epsilon \log T$$

$$\leq \sum_{S \neq S^*} D_S d \left(\max_i \mu_i + \epsilon\right) \log T$$

$$= O(\log T).$$

$\square$

**Proposition 3.6.** *Consider a CMAB problem with unbounded rewards and signed means. Assume the learner experience $O(\log T)$ regret, and the oracle attacker follows the attack strategy in Eq. (5). Then, if Condition 3.3 holds, the attack is successful i.e., $\mathbb{E}[N_{S^*}(T)] = T - o(T)$ with linear cost $O(T)$.*

*Proof.* The first claim of the proposition regarding the successfulness follows the same steps described in the first part of the proofs of Proposition 3.3. When the attacker corrupts also arms in the target $S^*$, the cost is:

$$\mathbb{E}\left[\sum_{t=1}^{T} |A_{S_t}^\epsilon|\right] = \mathbb{E}N_{S^*}(T) \cdot |A_{S^*}^\epsilon| + \sum_{S \neq S^*} \mathbb{E}N_S(T) \cdot |A_S^\epsilon|$$

$$\leq |A_{S^*}^\epsilon|T - |A_{S^*}^\epsilon| \log T + \sum_{S \neq S^*} D_S |A_S^\epsilon| \log T$$

$$\leq d(\max_{i \in [m]} \mu_i)T - d(\max_{i \in [m]} \mu_i) \log T + \sum_{S \neq S^*} D_S d \left(\max_{i \in [m]} \mu_i + \epsilon\right) \log T$$

$$\leq d(\max_{i \in [m]} \mu_i)T - d(\max_{i \in [m]} \mu_i) \log T + \sum_{S \neq S^*} D_S d \left(\max_{i \in [m]} \mu_i + \epsilon\right) \log T$$

$$= O(T). \tag{16}$$

$\square$

### A.3 REALISTIC ATTACKS

**Proposition 4.2.** *Consider a CMAB problem with unbounded rewards and positive means. Assume the learner experiences $O(\log T)$ regret, and the attacker follows the attack strategy in Eq. (8). Then, the attack is successful, i.e., $\mathbb{E}[N_{S^*}(T)] = T - o(T)$ with cost $O(\log T)$.*

*Proof.* First, we prove an upper bound for the number of target arm pulls $\mathbb{E}\{N_{S^*}\}$. Thanks to Lemma 4.1, we see with probability $1 - \delta$ that $\mu_i \leq \hat{\mu}_i(t) + \beta(N_i(t))$. Now consider $\hat{\mu}_i^c(t)$ as the

corrupted estimate for basic arm $i$ at time $t$ observed by the learner, then:

$$\hat{\mu}_i^c(t) = \hat{\mu}_i(t) - \frac{\sum_{\tau \in T_i(t-1)} (\hat{\mu}_i(\tau) + \epsilon)}{N_i(t-1)} \tag{17}$$

$$\leq \mu_i + \beta(N_i(t)) - \frac{\left(\sum_{\tau \in T_i(t-1)} \mu_i - \beta(\tau)\right)}{N_i(t-1)} - \epsilon \tag{18}$$

$$= \mu_i + \beta(N_i(t)) - \frac{\sum_{\tau \in T_i(t-1)} \mu_i}{N_i(t-1)} + \frac{\sum_{\tau \in T_i(t-1)} \beta(\tau)}{N_i(t-1)} - \varepsilon \tag{19}$$

$$= \beta(N_i(t)) + \frac{\sum_{\tau \in T_i(t-1)} \beta(\tau)}{N_i(t-1)} - \epsilon.$$

Where in Inequality 17 we separate the total corruption injected until round $t-1$. In Inequality 18 and Equation 19 we can extract $\epsilon$ remove $\mu_i$ noticing that $\sum_{\tau \in T_i(t-1)} k = k N_i(t-1)$. Now we can explicit the confidence radius $\beta(.)$ to obtain:

$$= \sqrt{\frac{\log\left(\frac{2mT}{\delta}\right) 2\sigma^2}{N_i(t)}} + \frac{\sum_{\tau \in T_i(t-1)} \sqrt{\frac{\log\left(\frac{2mT}{\delta} 2\sigma^2\right)}{N_i(\tau)}}}{N_i(t-1)} - \epsilon$$

$$= \sqrt{\frac{\log\left(\frac{2mT}{\delta}\right) 2\sigma^2}{N_i(t)}} + \sqrt{\log\left(\frac{2mT}{\delta} 2\sigma^2\right)} \cdot \frac{\sum_{\tau \in T_i(t-1)} \sqrt{\frac{1}{N_i(\tau)}}}{N_i(t-1)} - \epsilon \tag{20}$$

$$\leq \sqrt{\frac{\log\left(\frac{2mT}{\delta}\right) 2\sigma^2}{N_i(t)}} + \sqrt{\log\left(\frac{2mT}{\delta} 2\sigma^2\right)} O\left(\frac{1}{\sqrt{N_i(t-1)}}\right) - \epsilon \tag{21}$$

Where the term $\frac{\sum_{\tau \in T_i(t-1)} \sqrt{\frac{1}{N_i(\tau)}}}{N_i(t-1)}$ in Inequality 20 is a Puiseux series and the bound in Inequality 21 is obtained by the Euler-Maclaurin sum formula. Then, to obtain a bound over the number of pulls $N_i(t)$ for arm $i$, we reformulate the last 21 knowing that its value is negative as follows:

$$O\left(\frac{1}{\sqrt{N_i(t-1)}} \sqrt{\log\left(\frac{2mT}{\delta} 2\sigma^2\right)}\right) - \epsilon < 0 \tag{22}$$

Solving for $N_i(t-1)$ we obtain:

$$\Omega(\sqrt{N_i(t-1)}) > O\left(\frac{\sqrt{\log\left(\frac{2mT}{\delta} 2\sigma^2\right)}}{\epsilon}\right) \tag{23}$$

Then, summing over all basic arms that are not in the target $i \notin S^*$, we obtain the number of pulls of every arm $S \neq S^*$:

$$\sum_{S \neq S^*} N_S(t-1) \leq \sum_{i \notin S^*} O\left(\frac{\log\left(\frac{2mT}{\delta} 2\sigma^2\right)}{\epsilon}\right)$$

$$\leq d \cdot O\left(\frac{\log\left(\frac{2mT}{\delta} 2\sigma^2\right)}{\epsilon}\right)$$

The number of pulls for each arm $S \neq S^*$ at time $t$ is:

$$\sum_{S \neq S^*} N_S(t) \leq 1 + d \cdot O\left(\frac{\log\left(\frac{2mT}{\delta} 2\sigma^2\right)}{\epsilon}\right) \tag{24}$$

Finally, the upper bound for the pulls of the $S^*$ arm is then:

$$N_{S^*}(T) \geq T - \sum_{S \neq S^*} N_S(T)$$

$$\geq T - 1 - d \cdot O\left(\frac{\log\left(\frac{2mT}{\delta} 2\sigma^2\right)}{\epsilon}\right) \tag{25}$$

$\square$

**Proposition 4.3.** *Consider a CMAB problem with unbounded rewards and signed means. Assume the learner experiences $O(\log T)$ regret, and the attacker follows the attack strategy in Eq. 9 then the attack is successful i.e., $\mathbb{E}[N_{S^*}(T)] = T - o(T)$ with cost $O(T)$.*

*Proof.* The proof follow easily from 3.6, and Lemma 4.1. $\square$

## B    EXPERIMENTS

In this section, we provide minor details about the experiments omitted in the main paper.

**Experiments details**

- Experiment were conducted using `python 3.11.6`
- CPU: `Apple M1`
- RAM: `16 GB`
- Operating System: `macOS 14.2.1`
- System Type: `64 bit`

