# OpenReview forum: "Attacking Combinatorial Bandits: Beyond Bounded Rewards"
_ICLR.cc/2026/Conference — Submitted to ICLR 2026_

### Official Review · Reviewer_gEy2 · 2025-10-24

**Soundness:** 3
**Presentation:** 3
**Contribution:** 2
**Rating:** 4
**Confidence:** 4

**Summary:**

This paper designs attack policies for combinatorial bandits, considering three types of reward environments. For each environment, the paper identifies some concern cases where a sublinear attack is infeasible, and proposes attack policies for CUCB algorithms.

**Strengths:**

1. The writing of this paper is easy to follow.

**Weaknesses:**

Overall, the reviewer leans towards a mild negative evaluation of this paper, with the following concerns:

1. The most concerning issue of this paper is the novelty. The new parts of this paper are mainly Conditions 3.1, 3.2, and 3.3. However, from the reviewer’s perspective, these conditions can be straightforwardly derived from the CMAB’s standard structure without any technical challenges. If the authors are not agree with that, please add a discussion on the technical difficulties of deriving them.
2. For the attack on Condition 3, the authors claim that a linear budget is unavoidable. How about set the $\epsilon$ in Eq(3), as $\frac{1}{T}$ or $\frac{1}{\log T}$?
3. For the two attacking policies in Eq. (9) and Eq. (10), as in the realistic case, the attacker also does not know whether Condition 3.3 holds or not. How does it decide which attacking policy to pick?
4. In all the budget analyses, only the orders in terms of $T$ are given, leaving the dependence on gaps unclear. Given that the bandit literature is on the theoretical side, it would be better to derive and present fine-grained analysis.

**Questions:**

Please see the weaknesses above.

---

### Official Review · Reviewer_4PRb · 2025-10-26

**Soundness:** 2
**Presentation:** 2
**Contribution:** 2
**Rating:** 2
**Confidence:** 4

**Summary:**

This paper studies the problem of attacking combinatorial bandits under different conditions: 1) bounded rewards with positive means, 2) unbounded rewards with positive means and 3) unbounded rewards with signed means. It identifies necessary and sufficient structural conditions for attacks to be successful. Attack algorithms are given for the oracle case where the attacker knows the means of all the base arms, and the realistic case where it has the same information as the learner. The algorithms are proved sound and validated empirically on toy examples.

**Strengths:**

The paper formulates and solves an interesting problem of attacking combinatorial bandits where the rewards can be unbounded and may be negative. I appreciate the careful division into multiple cases and their analysis.

The analysis shows that unless some structural conditions hold, the bandits are not attackable. In the case where they are attackable, specific attack algorithms are given and are analyzed.

The empirical results validate the correctness of the algorithms.

**Weaknesses:**

Despite the above strengths, There are some problems with the specific claims and proofs in the paper. On the whole the paper needs careful reworking. Also the empirical results will be more impressive on less toy-like problems.

Confusing notations.
Algorithm 1. Clarify that Xs_t and As_t are vectors and Xs_t + As_t is a vector addition (not sets or numbers).
But then, below the Algorithm 1 you write As_t = \sum_i Ai_t and C(T) = \sum As_t. This does not make sense unless As_t is a number.

C2 is convoluted. it is better stated as"There exists a superarm S \subset S* s.t. \forall j \in S*/S, \mu_j =0."
However, I think that this needs to be generalized to cover all negative cases as I show below.

In the proof of Proposition 3.3. the definition of S^{1,2} is not proper. I think you mean (S*\S) \bigcap {i | \mu{i) > 0} =/= \Phi, i.e.,
For all S, S*\S has at least one arm with non-zero mu. If this is the case, when the adversary sets all arms in S/S* to have zero means, S* would be the best arm and will be chosen.  I suggest that you add this explanation to the main text.

However S^{1,2} is not the same as [ \not C1 \wedge \not C2]. Suppose C1 and C2 are both false. In particular assume that there is no S where S subset S* or S* \subset S.  Let S* = {a, b} S ={b,c} and mu(a)=0. The algorithm does not seem to work, since setting mu(c) to 0, would keep both S and S* to be optimal.

One way to fix this is to make C2 weaker. Perhaps: "There exists a superarm S, where S*/ S is non-empty and \forall j \in S*/S, \mu_j =0." , i.e., S need not be a strict subset of S*.

In this case S^{1,2} seems equal to [\not C1 \wedge \not C2] and the proofs seem to go through.

Equation (8) I suspect  that A_{i,t} should be set to -(\hat{\mu}_i(t)+\beta(N_i(t)+\epsilon) if i \in S_t\S*.

The reason is that \hat{\mu} might have statistical error and the real \mu may be higher by \beta. Equation 9 does this for example.

Page 14. Proof of Proposition 3.6. The same equation is repeated twice. It appears that epsilon should come in the first part of the repeated equation and not on the second part.

Page 15, Proof of Proposition 4.2. < sign in Eq 23 is turned into >= in Eq 24. Why?
It appears that equation is squared on both sides and summed, but if so \epsilon should have turned into epsilon squared. This step should be explained. As I said earlier, I think that Equation (8) is wrong. If so the proof would be wrong too.

In Proposition 4.3. N_{i*}(t) is not defined. Equation 10 should be explained and Proposition 4.4 should be proved.

Also, Lemma 4.1. claims that the expectation of an event E is > 1-\delta. Your propositions that rely on 4.1 should include this caveat (only true with probability 1-\delta.

Typo level comments.
Line 183. "be addressED"
Line 220. \mu* and \mu*_S* are both used. Are they the same?
L 228. "this conditions" -> "these conditions"
L 241. "null mean" -> "zero mean"
L 365. "begins the after" -> "begins after"
L 376. "dependent by" -> "dependent on"

**Questions:**

1. Answer my question about the correctness of C2 and S^{1,2} above.

2. Is Equation 8 correct? What about the proof of proposition 4.2.?

3. Explain Equation 10 and justify Proposition 4.4.

---

### Official Review · Reviewer_riyv · 2025-11-01

**Soundness:** 3
**Presentation:** 2
**Contribution:** 2
**Rating:** 2
**Confidence:** 4

**Summary:**

The paper studies adversarial attacks on combinatorial multi-armed bandits (CMAB) under a linear reward model. It formalizes attack cost and impact trade-offs under different conditions and proposes an attack strategy designed to degrade the performance of a CUCB learner. The theoretical analysis provides regret and cost bounds under several variants of attack settings. The authors also present numerical results to illustrate the attack’s effectiveness.

**Strengths:**

1) The study of adversarial attacks in bandit settings continues to be relevant, especially for complex decision-making environments such as CMABs.
2) The paper systematically presents its assumptions, theoretical analysis, and experimental validation across different CMAB attack settings.
3) The experimental results, though limited in scope, effectively illustrate the theoretical findings and demonstrate the attack’s impact under the assumed linear reward setting.

**Weaknesses:**

1) The entire analysis relies on the assumption of a linear reward model. While this simplifies analysis, it restricts the generality of the results. The linear model is not representative of general CMAB settings, where nonlinear or triggering-based reward dependencies are common. It is unclear whether the techniques can extend to non-linear or triggered CMAB models.
2) Given the linear reward assumption, the results in Section 3 seem expected and primarily incremental. The theoretical findings do not appear to introduce fundamentally new challenges beyond adapting existing bandit-attack analyses (e.g., [Jun et al., 2018]) to the CMAB domain.
3) The discussion of linear attack costs is somewhat trivial. It offers limited insight into more realistic attack models where costs are non-linear or constrained.
4. The attack proposed in Section 4 appears conceptually similar to the general attack framework of [Liu et al., 2019]. It would be valuable to investigate whether a more tailored attack strategy could be developed for CUCB, analogous to the specialized designs for UCB in [Jun et al., 2018].

**Questions:**

1) Can the authors clarify how the linear reward assumption affects generality? Can the attack design be adapted to specific CMAB structures (e.g., matching, subset selection) to show the generality of the approach?
2) What are the main technical challenges compared to prior attack analyses in [Jun et al., 2018] or [Liu et al., 2019]?

---

### Meta-Review · Area_Chair_A1E1 · 2026-01-04

**Summary:**

Multiple serious concerns have been raised regarding this paper, including potential errors in specific claims and proofs. Furthermore, reviewers pointed out the limited scope of the proposed applications and raised significant doubts about the technical novelty of the work. Despite the breadth and severity of these critiques, the authors did not submit a rebuttal, leaving all identified issues completely unaddressed.

**Reviewer Concerns:**

Addressed by Rebuttal: None. The authors did not participate in the rebuttal process.

Outstanding Concerns: All concerns remain outstanding. These include:

- Potential technical errors in the core claims and mathematical proofs.

- The limited applicability and narrow scope of the proposed method.

- A lack of demonstrated technical novelty relative to existing work.

- General lack of clarity in the presentation.

**Reviewer Scores:**

Since no rebuttal or additional clarification was provided by the authors, I predict that the scores would have remained unchanged. In the absence of any response to the technical doubts and criticisms raised during the review phase, there is no basis for the reviewers to revise their initial negative assessments. Had a discussion period occurred, the lack of author engagement would have only confirmed the initial concerns, likely leading to a consensus for rejection.

---

### Decision · Program_Chairs · 2026-01-26

Reject